# Multiplexing Quadrupole and Ion Trap Operation Modes on a “Brick” Miniature Mass Spectrometer

**DOI:** 10.3390/molecules28227640

**Published:** 2023-11-17

**Authors:** Chaohong Feng, Siyu Liu, Ting Jiang, Wei Xu

**Affiliations:** School of Medical Technology, Beijing Institute of Technology, Beijing 100081, China; 3120211461@bit.edu.cn (C.F.); m18810932050@163.com (S.L.)

**Keywords:** ion trap, quadrupole, miniature mass spectrometer, PCB-based multi-electrode mass analyzer

## Abstract

Although a quadruple mass analyzer and an ion trap mass analyzer have complementary analytical features, they usually have different geometries, operational modes, and electronic control systems. As a continuous effort to extend its coverage, both quadrupole and ion trap operation modes were realized on a “brick” miniature mass spectrometer with a single mass analyzer. In the quadrupole operation mode, low-mass ions ranging from 31 to 502 Th can be analyzed. On the other hand, the ion trap mode can be utilized to cover ions with higher mass to charge ratios (up to 922 Th), as well as performing tandem mass spectrometry. To realize the multiplexing of both operation modes, a printed circuit board (PCB)-based multi-electrode quadrupole–ion trap mass analyzer was designed and integrated in the system. To cover both volatile and non-volatile molecules, two ionization sources were also implemented, including a nano electrospray ionization source and an in-vacuum plasma ionization source. Performances of the instrument operated in these two modes were characterized, such as mass resolution, sensitivity, and mass range. Results demonstrate that the combination of the quadrupole and ion trap operation modes can provide new capabilities when solving analytical problems.

## 1. Introduction

Owing to its high portability and satisfactory analytical capabilities [1,2,3], miniature mass spectrometry (MS) has extended its applications into different areas, including environmental monitoring, food safety, and point-of-care testing [4,5]. In recent decades, significant efforts have been made toward the miniaturization of an MS device. The advancements in the discontinuous atmospheric pressure interface (CAPI) [6,7] and the continuous atmospheric pressure interface (DAPI) [8,9,10,11] have facilitated the analysis of volatile and nonvolatile samples using compact vacuum systems. Various types of ion traps have been designed and miniaturized, such as cylindrical ion traps (CITs) [12], linear ion traps (LITs) [13,14], and rectangular ion traps (RITs) [15]. In addition to the traditional method of radio frequency (rf) amplitude scanning, frequency scanning technique has also been implemented as an alternate approach for driving ion traps, making MS systems to be even more compact [16].

After obtaining these platforms, additional attention has been given in the improvement of analytical performance in terms of sensitivity, MS resolution, stability, quantitation accuracy, and mass range. A miniature ion funnel was incorporated into miniature mass spectrometers to achieve sub-ppb level detection of limit (LOD) and unit mass resolution. Compact multistage vacuum system coupled with CAPI was developed to allow high repeatability and stability analysis [17,18]. Pseudo-SRM and pseudo-MRM modes were realized on a miniature mass spectrometer using the Grid-SWIFT waveform during the ion introduction period, so that higher detection sensitivity and quantitation accuracy can be achieved for target analysis [19,20]. Miniature ion trap mass spectrometers typically have a mass range of 100–2000 Th. To increase the mass coverage for high-molecule analysis, methods from the perspectives of novel operation modes and modified hardware were explored. Frequency scanning techniques such as digital waveforms and sinusoidal waveforms were used to drive ion traps [21,22,23,24,25]. More recently, hybrid ion scan mode was developed to allow the analysis of both small and large molecules from 100 to 12,000 Th [26,27]. However, in order to evaluate volatile samples less than 100 Th, the detection of high-mass ions must be compromised. By increasing the radius of the ion trap, a mass range of 50–400 Th can be achieved on a “brick” miniature ion trap mass spectrometer. For lighter ions below 50 Th, further voltage reduction and an expansion of the ion trap radius were required. Ion traps, however, present limitations in terms of low mass coverage as a result of the variations in trapping efficiency. Analyzing low-mass ions without compromising high-mass ion analysis on the same MS platform was still challenged.

Similar to ion traps, quadrupole mass analyzers (or quadrupole mass filters, QMFs) provide several advantages, including straightforward design and cost-effectiveness, which contribute to their widespread use in MS systems [28,29,30]. There are two working modes for quadrupole mass analyzers: the ion guide mode for ion transfer and the mass filter mode for mass analysis. In the ion guide mode, ions with a wide range of m/z values have stable trajectories and can pass through the quadrupole. In the mass filter mode, only ions with specific m/z values can selectively reach the detector [31]. Thereby, the space charge effects can be eliminated, and the quantitative accuracy can be improved. In order to obtain a complete mass scan, the direct current (DC) and radio frequency (rf) voltages applied to the electrodes need to be continuously varied while maintaining a constant ratio [32]. When analyzing large molecules, the required voltages can reach several thousand or even tens of thousands of volts. This introduces complexities in the circuit, including resonance point drift under high voltage and potential discharge issues within the device [33]. Mathieu’s equation demonstrates that ion mass is also related to frequency. By employing a frequency scanning mode where the DC and rf maintain a constant and the frequency is swept from high to low, it is possible to analyze ions with high m/z values using lower rf voltages. The frequency scanning mode simplifies the circuit and expands the capabilities of mass spectrometry for exploring a wider range of masses. Quadrupoles used in benchtop GC-MS are typically operated in the mass range of 10–1000 Th [34]. Quantitative analysis of complex samples was usually performed on quadrupole MS [35,36,37]. However, a single quadrupole cannot achieve tandem mass spectrometry and requires a tandem-in-space configuration to enable MS/MS experiments.

Previously, miniature ion trap mass spectrometers driven by a frequency scanning technique were developed [6,24,27,38,39]. As a continuous effort to extend its mass coverage, quadrupole and ion trap operation modes were multiplexed. A compact three-stage vacuum chamber with a pressure control valve was developed, enabling pressure control between quadrupole and ion trap operation modes. A PCB-based multi-electrode quadrupole–ion trap mass analyzer was designed and integrated in the system. Ion sources of an in-vacuum plasma ionization and a nanoESI were implemented so that volatile and nonvolatile samples can be ionized. The uniqueness of the instrument is that the same mass analyzer can be run in two different modes. In the quadrupole operation mode, low-mass ions ranging from 20 to 600 Th can be analyzed. On the other hand, the ion trap operation mode was used to analyze ions with higher mass to charge ratios (up to 1000 Th) as well as performing tandem MS. Instrument performance of mass resolution, sensitivity, and mass range in these two modes was characterized.

## 2. Results and Discussions

### 2.1. Design of a PCB-Based Multi-Electrode Quadrupole–Ion Trap Mass Analyzer

To realize the combination of the quadrupole and ion trap operation modes on a single instrument, a PCB-based multi-electrode quadrupole–ion trap mass analyzer was designed. As shown in Figure 1b, the PCB-based mass analyzer consists of the rf electrode, the guide electrode, and the alternating current (AC) electrode. Both rf and AC electrodes were assembled using four mortise and tenon structures with a dimension of 5 mm in length and 1.6 mm in width. This joint structure provides a coarse alignment between the electrodes. In order to finely control the electrode position, two stainless-steel holders were utilized to connect both ends of the PCB. Two stainless-steel end-cap electrodes with a 3 mm diameter aperture were soldered onto the PEEK holders.

Previous studies have demonstrated that the coupling of high-order fields to the quadrupole field can effectively enhance the performance of quadrupole mass filters and ion traps [40,41,42]. As a demonstration, QMFs with different dimension from QMF-1 to QMF-8 were designed. High-order fields, including the quadrupole field (A2), the hexapole field (A3), and the octupole field (A4), were calculated using the least-square fitting method. The components of high-order fields in different QMFs are shown in Appendix A. The nonideal electric fields were imported into the ion trajectory simulation program to calculate the MS resolution and ion transmission ratios. As indicated in Appendix A, the MS resolution and ion transmission ratios of the quadrupole mass filter were observed as an interdependent relationship. The higher MS resolution of quadrupole mass filters can be achieved at the cost of sacrificing the sensitivity. As a trade-off, QMF-8 with a hexapole field of −1.45% and octupole field of 1.56% was manufactured and tested in the experiments.

Figure 1c shows the geometry and electric fields in the x–y plane when the PCB-based mass analyzer operated in the quadrupole mode. After optimization, PCB-based rf electrodes were fabricated with dimensions of 92 mm in length, 2.5 mm in width (w2), and 0.1 mm in thickness. Additionally, a segmented ion guide electrode was designed to transfer ions toward the end of the PCB-based mass analyzer. Following the optimization process, a width 0.5 mm (w1) was chosen for each segment, while the majority of segments were built with a length of 9 mm. A spacing of 0.8 mm (d1) was maintained between the AC and guiding electrodes. Guiding electrodes were interconnected by means of resistors. By applying voltage to the first and final segments of the guide electrodes, a gradient electric field can be introduced. In experiments, the mass analyzer with a dimension of 5.2 mm (rx) × 4 mm (ry) was adopted. Since the ion trap was asymmetric, the effective field radius was determined following the methodology proposed by Xu et al. [43]. The ion secular frequency was obtained by the Fourier transform of ion trajectory within ion traps. An effective radius of 4.7 mm was derived.

The presence of higher-order fields introduces deformations to the stability region [44,45]. Numerical simulations were also carried out to map the stability diagram with the presence of high-order fields. The fourth-order Runge–Kutta algorithm was used to calculate the trajectories of ion motion within the trap [46]. The initial ion distribution with 10^4^ ions was confined to an area of 0.04 mm^2^ centered at z = 0, with average energies of 0.3 eV, 0.3 eV, and 3 eV in the x, y, and z directions, respectively. Nitrogen was used as the buffer gas with a buffer gas pressure of 0.1 mTorr, and the Langevin collision model was applied to simulate the ion-neutral collisions. By scanning the rf, a scan of each (a, q) pair within the first stability region was iterated and recorded as shown in Figure 2a. In comparison with the ideal quadrupole field, the apex shifted from (0.707, 0.236) to (0.730, 0.245). The enlarged first stability region was found to be beneficial impacts in terms of expanding the capacity for low-mass detection and enhancing the MS resolution.

### 2.2. Multiplexing Quadrupole and Ion Trap Operation Modes

Similar to a conventional ion trap, the scan function of the ion trap driven by a frequency scanning technique also has three periods: injection, cooling, and MS scan [24]. The three periods last with durations of 100 ms, 100 ms, and 200 ms, respectively. The scheme of the experimental sequence operated in the ion trap mode is shown in Figure 2c. The voltage of the rear-end cap (DC2) was fixed at +100 V. The front-end cap (DC1) voltage was first set at 0 V during the ion injection periods, then fixed at 100 V, allowing for ion cooling and MS scan. Segmented guide electrodes were designed to increase the ion transport efficiency and let most ions be trapped in the final section of the ion trap. The voltage applied to the first segment was set at 0 V, and the last segment of the guide electrodes was optimized, as shown in Appendix A. Results indicate that when the voltage of the end guide was set at −25 V, a maximum ion abundance can be obtained, which is 3-fold higher than other voltages. In the injection and cooling periods, the rf voltage amplitude and frequency were kept constant for ion trapping. During MS scan periods, the rf amplitude remained constant at 500 Vp-p, while their frequencies swept from 1 MHz to 300 kHz. Additionally, the AC frequency varied from 333 kHz to 133 kHz, with an amplitude of 1.2 Vp-p. In this way, a resonance ejection point value of q = 0.784 was realized for ion ejection.

In contrast to the ion trap mode, the quadrupole is operated with a continuous beam of ions with various m/z values, wherein the rf is ramped to allow specific ions to pass through the stability region. The principle of the frequency scanning quadrupole is plotted as shown in Figure 2d. During the MS scan, the voltage of U applied on the AC electrode was kept at 12.6 V. The rf was varied while keeping the ratio between the DC and rf voltage constant so that a straight scan line was obtained. The performance of the quadrupole mass filter is determined by the scan line. When the scan line is placed at a higher level, it results in a decrease in sensitivity. Conversely, in the opposite scenario, the resolution of the MS is compromised. As a trade-off, the scan line in the quadrupole operation mode was set at 0.168. During the periods of MS scanning, the frequency of rf swept from 1.2 MHz to 0.2 MHz with an amplitude of 150 Vp-p. A lower mass limit of 20 Th was achieved. Figure 2b illustrated the mass scanning diagrams in the quadrupole mode, which was operated using a sinusoidal waveform frequency scanning technique, within the f-U space. Ions with lower m/z values will initially pass through the apex of the stability diagram, while other ions will rapidly become unstable and strike the electrodes. As the frequencies swept, ions with higher m/z values would pass through. In this way, a mass spectrum is obtained for ions with mass to charge values ranging from small to large.

To achieve a linear mass calibration, a nonlinear frequency scan was used. The relationship between frequency (f) and the scan time (t) is expressed as the start frequency and end frequency described as follows:(1)$$f=\sqrt{\frac{1}{\frac{1}{T}(\frac{1}{f_{end}^{2}}-\frac{1}{f_{start}^{2}})t+\frac{1}{f_{start}^{2}}}}$$

According to the Mathieu equation, the relationship of m/z and rf was described using two reduced parameters, a and q:(2)$$a=\frac{8zeU}{mr_{0}^{2}\Omega^{2}}q=\frac{4zeV}{mr_{0}^{2}\Omega^{2}}$$
where V is the amplitude of the rf, z is the number of charges the ion possesses, and r_0_ is the effective radius. T is the total scan time, and U is the DC voltage applied on a pair of electrodes. In a typical voltage scanning quadrupole, the rf amplitude and DC voltage need to be scanned with a fixed ratio of λ = 2a/q = U/V. In a frequency scanning quadrupole, the rf amplitude and DC voltage are kept constant. By substituting Equation (3) into Equation (2) using f = Ω/2π, a linear m/z relationship related to time can be derived. When operating in the ion trap mode, q = q_eiect_ represents the ejection point of q value using a resonance ejection.
(3)$$\frac{m}{z}=\frac{V}{qr_{0}^{2}\pi^{2}f_{start}^{2}}+\frac{V}{qr_{0}^{2}\pi^{2}}\left(\frac{1}{f_{end}^{2}}-\frac{1}{f_{start}^{2}}\right)\cdot \frac{t}{T}$$

### 2.3. Characterization of the Quadrupole Operation Mode

#### 2.3.1. Mass Resolution

In general, the quadrupole mass filter operates at pressures below 10^−5^ Torr. Elevated buffer gas pressure would reduce the kinetic energy along the ion’s axial motion. As a result, the ions are unable to reach the detector located at the end of the analyzer. When working in the quadrupole operation mode, the pressure in the three-stage chamber was maintained at ~0.08 mTorr. To characterize the MS resolution in the quadrupole mode, the scan rate was modified by increasing the scan time. Methanol was used as a sample to test. As illustrated in Figure 3a, the FWHM of ~2 Th was observed at the scan rate of 3200 Th/s. The MS resolution can be improved by lowering the MS scan rate. Better than unit resolution (0.5 Th) can be achieved when decreasing the MS scan rate to 640 Th/s.

#### 2.3.2. Sensitivity

Under optimized conditions, the sensitivity of the instrument operated in the quadrupole mode was characterized. The capability of the quadrupole mode coupled with in-vacuum plasma ionization source analysis for volatile organic compounds was demonstrated. The headspace vapor pressure technique was applied to prepare gas samples with certain concentrations. Ethanol was prepared in a glass bottle sealed with a rubber stopper for hours to achieve saturated vapor pressure. The saturated ethanol was extracted from the bottle using a syringe and then pumped into the MS inlet through a polytetrafluoroethylene (PTFE) tube (1/4 inch outer diameter, 1 m length). Detailed information about this quantitative method can be found in our previous work [7]. Figure 3b presents the linear range for ethanol detection. A limit of detection (LOD) of 100 ppb was achieved. A good linearity (R^2^ > 0.999) of the quantitation curve was observed within the range of 100 ppb to 5000 ppb. A good quantitative accuracy can be achieved with the capability of eliminating interference ions in real time. The relative standard deviations (RSDs) were calculated for ethanol. An RSD of better than 3.5% was achieved over a concentration range of 100−5000 ng/mL as shown in Table 1.

#### 2.3.3. Mass Range

Besides MS resolution and sensitivity, mass coverage is another performance required to be characterized. As a demonstration, perfluorotributylamine (PFTBA) was first tested. The mass spectrum of perfluorotributylamine (PFTBA) covers a mass range from 31 to 502 Th, and some fragments can be observed, as shown in Figure 3c. Additionally, several other VOCs were also validated using the quadrupole mode. Figure 4 shows representative mass spectra collected from the quadrupole operation mode using the in-vacuum plasma ionization source. Either molecular ions or protonated ions were produced and observed in their mass spectra. Protonated monomers and dimers were observed in methanol (m/z 32) and ethyl acetate (m/z 88), as presented in Figure 4a,d. With different chemical structures and properties, xylene and ethanol were identified as molecular ion peaks (M^+^). Additionally, fragmented ions at m/z 92 for xylene and m/z 32 for ethanol were also observed in mass spectra shown in Figure 4c,b. In experiments, vials containing the solid or liquid samples were placed below the MS inlet to let it vaporize into the MS for direct analysis. As reported in a previous work, ion fragmentations were affected by the temperature and the power of the plasma ionization source [7].

### 2.4. Characterization of the Ion Trap Operation Mode

#### 2.4.1. Mass Resolution

The resolution of an ion trap is affected by the buffer gas pressure in it. Elevated pressures can help cool ions to the center of the ion trap, but it would also cause mass peak broadening. In the ion trap operation mode, the buffer gas pressure in an ion trap was regulated at ~1.5 mTorr by using a valve so that a high mass resolution and ion capture efficiency can be achieved. To characterize the MS resolution, the FWHM at different scan rates was recorded and plotted in Figure 5a. A sample solution of imatinib (m/z 494) with a concentration of 1 μg/mL was ionized by a nano-ESI source. An FWHM of ~3.71 was obtained at a scan rate of 9000 Th/s. A unit mass resolution was achieved when decreasing the scan rate to 2250 Th/s. The mass resolution can be further improved by lowering the scan rates. However, ion intensity would be reduced as the scan rate was too low. Therefore, optimization of the scan rate needs to be performed to balance the mass resolution and sensitivity. In the ion trap operation mode, a typical scan rate of 4000 Th/s can be used with a mass resolution of 1.4 Th.

#### 2.4.2. Mass Range

Compared with the conventional voltage scanning method, a broader mass range can be achieved using a frequency scanning approach at the same rf amplitude. In this experiment, the rf was swept from 1 MHz to 300 kHz with an amplitude of 450 V_p-p_, and a mass range of 100–1000 Da was obtained. Figure 5b plots the mass spectrum of the Tuning Mix, which covers the mass range from 322 to 922 Da. Ions with high m/z values can also be detected by further reducing the frequency and increasing the amplitude of the rf as described in our previous research.

#### 2.4.3. Sensitivity

The resolution of an ion trap is affected by the buffer gas pressure in it. Elevated pressures can help cool ions to the center of the ion trap. The ion accumulation capability would enhance the sensitivity of an MS system, especially for detecting low-abundant ions. With the benefits of ion accumulation, the sensitivity in the ion trap operating mode would be expected to be much higher than that in the quadrupole operation mode. As a demonstration, experiments were carried out using imatinib as a testing sample to characterize the sensitivity of the instrument when operated in the ion trap mode. Imatinib solutions with a concentration from 10 ng/mL to 1000 ng/mL were analyzed using nanoESI as the ionization source. Figure 5c plots the linear range of detection of imatinib. A LOD of 10 ng/mL was achieved with good linearities of R^2^ > 0.99. Figure 5d plots the tandem mass spectrum of imatinib at 10 ng/mL. As expected, the concentration of analytes identified by the ion trap mode was lower than that detected by the quadrupole mode on the same platform. However, the quantification accuracy of the ion trap mode was inferior to the quadrupole mode. The RSDs of imatinib in the ion trap mode were also evaluated. An RSD of better than 8.3% was achieved over a concentration range of 10−1000 ng/mL, as shown in Table 2. In comparison, a smaller RSD (<8.3% vs. <3.5%) was obtained when the instrument operated in the quadrupole mode. With the capability of trapping and selecting ions, it is possible to modify this design to conduct gas-phase reactions. By altering the polarity of the transfer voltage, the generated positive and negative ions can be guided into the ion trap, respectively. Furthermore, altering the voltage applied on the guide electrode, as well as the voltages of the front and rear plates, also allows for precise control over the process of the reaction of these positive and negative ions.

## 3. Instrumental Section

### 3.1. Chemical Samples

Imatinib (MW 493.6) was purchased from Sigma-Aldrich (St. Louis, MO, USA). Tuning Mix (utilized for ion trap experiments) was obtained from Agilent (Palo Alto, CA, USA). Methanol (MW 32.04) was acquired from Dikmapure (Beijing, China), while ethanol (MW 46.06) was purchased from JDTZ (Tianjin, China). Perfluorotributylamine (MW 671.09) was purchased from Alta Scientific Co. (Tianjin, China). Dimethylbenzene (MW 106.16) was purchased from Meryer (Shanghai, China), and ethyl acetate (with m/z 88.11) was obtained from J&K Scientific (Beijing, China). Deionized water was acquired from Wahaha Co. (Hangzhou, China). All experimental procedures were conducted within a laboratory environment maintained at a constant temperature of 25 °C.

### 3.2. Instrumentation

Experiments were performed on a newly developed miniature mass spectrometer, which consisted of a compact three-stage vacuum system, a continuous pressure interface (CAPI), a miniature ion funnel, a quadrupole ion guide, a PCB mass analyzer, two detectors, and electronics, as shown in Figure 1a. The three-stage vacuums were driven by a combination of a scroll pump (SVF-E0-50, ScrollTech. Inc., Hangzhou, China) and two turbo pumps (Hipace10, Pfeiffer Inc., Asslar, Germany). A stainless-steel capillary (0.38 mm i.d., 10 cm in length) served as the sample inlet, connecting the atmosphere with the first chamber. The first and second chambers were connected through a skimmer (0.5 mm i.d.). A pinhole with a diameter of 3 mm was employed to establish a connection between the second and third chambers. To enhance ion transmission efficiencies, a previously reported miniature ion funnel was integrated into the first chamber [18], while a quadrupole ion guide was positioned in the second chamber. A PCB-based mass analyzer was placed in the third chamber for mass analysis, which enabled the multiplexing of a quadrupole mass filter and an ion trap (see Section 3.1). As shown in Appendix A, the whole of the system (including the vacuum chambers) has an approximate size of 355 mm in length, 275 mm in width, and 317 mm in height. The pressures in the first two chambers were maintained at ~5 Torr and ~6 mTorr, respectively. A metering valve (1/4 inch, Xiongchuan Valve Co., Beijing, China) was utilized for precise pressure control within the third chamber, enabling the quadrupole mass filter and ion trap to operate at their respective optimized pressures. In the ion trap operation mode, the pressure was tuned to ~2 mTorr, while in the quadrupole mode, the pressure was maintained at ~0.1 mTorr. Two detectors were positioned in the third chamber, with one in the axial direction and the other in the radial direction. A dual-source configuration with an in-vacuum plasma and nanoESI was implemented on the miniature mass spectrometers to expand the analyte coverage. The in-vacuum plasma ionization source, placed in front of the ion funnel, consisted of two stainless ring electrodes (I.D. 5 mm, thickness 0.5 mm). The spacing between the two electrodes was 2 mm. To generate the ionization plasma plume, a DC voltage of ~500 V generated by a high-voltage power supply (TianJin Dongwen High Voltage Power Supply Co., Ltd., Tianjin, China) was applied between the two electrodes as the discharge voltage. By turning on or off the high voltage applied on the ring electrode of the in-vacuum ionization source and adjusting the distance between the capillary and the first electrode of the ion funnel, a rapid switching between two ion sources was realized.

## 4. Conclusions

In this study, a compact three-stage vacuum system was developed to achieve lower pressures. By altering the pressure within the third vacuum chamber, multiplexing the quadrupole and ion trap operation modes was realized on this “brick” miniature mass spectrometer. The high-order fields of a PCB-based analyzer were optimized via ion trajectory simulation, and the stability diagram with nonideal electric fields was calculated. To characterize the two operation modes, two ion sources of a plasma ionization source and a nanoESI source were integrated into the instrument. For small molecules, such as VOCs, both qualitative and quantitative analyses were carried out in the quadrupole operation mode. An FWHM of 0.5 Th can be achieved for ethanol at a scan rate of 640 Th/s. A good linear detection range can be obtained from 100 ppb to 5000 ppb. A low-mass end of 31 Th can be reached with a high-mass end of 502 Th. Furthermore, in the case of nonvolatile samples such as drugs and peptides, the ion trap mode can provide high sensitivity and additional structural information. The performance in the ion trap mode was also assessed. A LOD of 10 ppb was achieved for imatinib, followed by the implementation of the tandem MS technique. Combining quadrupole and ion trap modes on a single instrument presents new capabilities for solving analytical problems. The employment of the quadrupole mode enables the analysis of low-mass molecules, while the ion trap mode facilitates sensitive and confirmatory analysis.

## Figures and Tables

**Figure 1 molecules-28-07640-f001:**
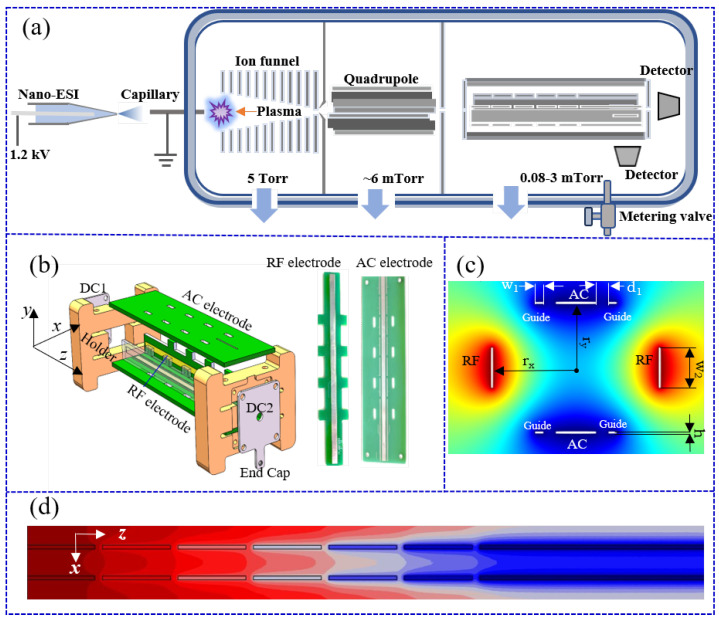
(**a**) Schematic diagram of the three-stage vacuum system. (**b**) An assembled prototype of PCB-based mass analyzer and pictures of rf and AC electrode (from left to right). The electric field distribution on the (**c**) x−y plane (z = 0), and (**d**) x–z plane in the ion trap operation mode.

**Figure 2 molecules-28-07640-f002:**
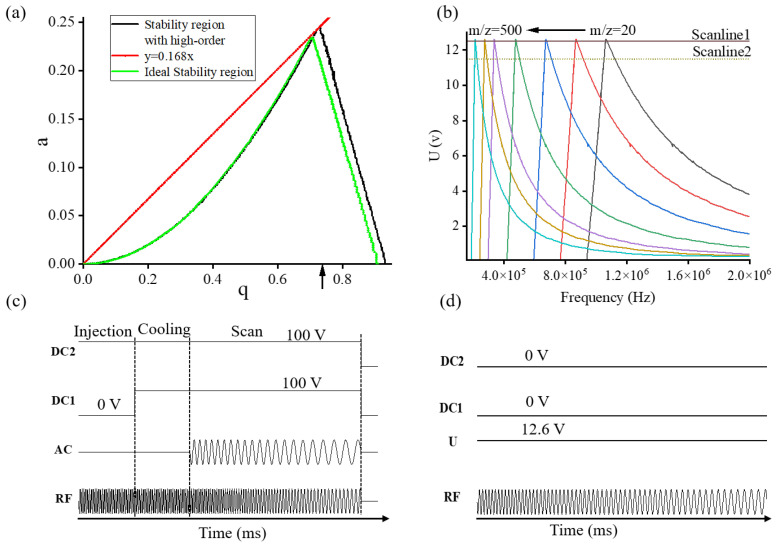
(**a**) A comparison of ideal and nonideal first stability regions with high-order fields. (**b**) Mass scanning diagrams in the quadrupole mode using sinusoidal waveform frequency scanning technique plotted in the f-U space. Scheme of the experimental sequence in (**c**) ion trap and (**d**) quadrupole operation modes.

**Figure 3 molecules-28-07640-f003:**
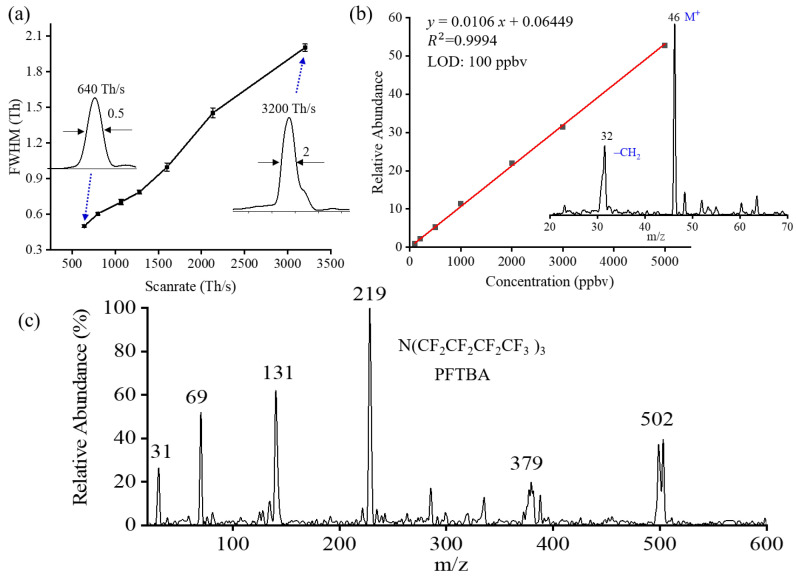
Quadrupole mass filter mode. (**a**) The FWHM of different scan rates (methanol). (**b**) Linear range of detection for ethanol. (**c**) The mass spectrum of perfluorotributylamine (PFTBA).

**Figure 4 molecules-28-07640-f004:**
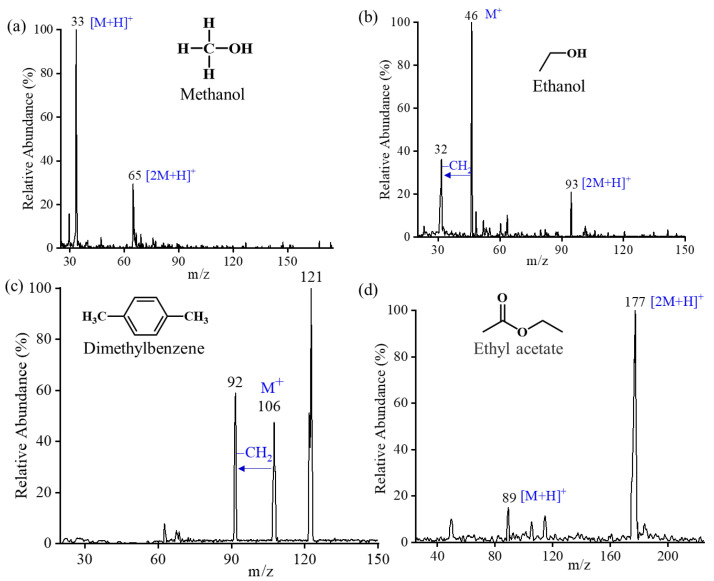
The mass spectra of (**a**) methanol (m/z 32), (**b**) ethanol (m/z 46), (**c**) dimethylbenzene (m/z 106), and (**d**) ethyl acetate (m/z 88).

**Figure 5 molecules-28-07640-f005:**
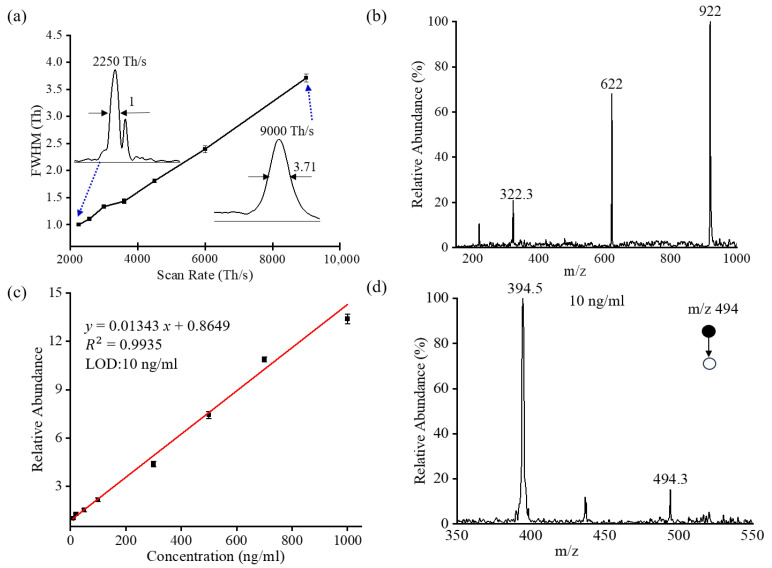
Ion trap operation mode. (**a**) The FWHM at different scan rates using imatinib as a sample. (**b**) The mass spectrum of Tunning Mix. (**c**) Linear range of detection for imatinib. (**d**) The tandem mass spectrum of imatinib (10 ng/mL).

**Table 1 molecules-28-07640-t001:** Relative Standard Deviations (RSDs) for Methanol Concentration Measurement in the Quadrupole Operation Mode.

Concentration (ng/mL)	100	200	500	1000	2000	3000	5000
RSD (%)	3.46	1.87	2.37	1.11	0.29	0.33	0.17

**Table 2 molecules-28-07640-t002:** Relative Standard Deviations (RSDs) for Imatinib Concentration Measurement in the Ion Trap Operation Mode.

Concentration (ng/mL)	10	20	50	100	300	500	700	1000
RSD (%)	5.48	8.31	7.18	5.35	3.60	2.97	1.41	2.30

## Data Availability

The data can be shared up on request.

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
