# Peer review of "Multiplexing Quadrupole and Ion Trap Operation Modes on a “Brick” Miniature Mass Spectrometer"

_molecules, 2023, doi:10.3390/molecules28227640_

Round 1

Reviewer 1 Report

Comments and Suggestions for Authors

Summary:

The authors set up a new experimental apparatus by incorporating quadruple mass analyzer and ion trap into a “brick” miniature mass spectrometer. The performance of this experimental setup in two operation modes is evaluated.

Impact:

The authors presented a novel instrumental design, which may be of interest to the readers of Molecules, especially for researchers in physical chemistry, spectroscopy and instrumental design. However, the impact of this design remains uncertain due to the following issues.

Issues:

1. The authors named this design as “miniature mass spectrometer”, what is the approximate size of the whole instrument? (Including the vacuum chambers)

2. In most cases, ion traps are used to conduct more complex research like cold atomic systems or gas phase reactions. What about the extensibility of this design? Will it be easy to modify that into a cold trap or reaction chamber? This should be explored in the discussion.

3. As the authors said in the introduction, quadruple mass analyzer and ion trap are different in many ways. Currently, to use the features from both, most research groups will stack quadruple mass analyzers and ion traps together. (i.e. ion guide + ion trap + ion guide) I appreciate the idea that a design can incorporate the features of both. However, the sizes of quadruple ion guide/filter and ion trap are not so big and a stack of both can usually fit inside a single vacuum chamber while having a much better mass resolution and extensibility. What are the unique advantages of this design?

Reviewer 2 Report

Comments and Suggestions for Authors

A printed circuit board (PCB)-based multi-electrode quadrupole-ion trap mass analyzer was designed and integrated into a “Brick” Miniature Mass Spectrometer, and realized the multiplexing of both quadrupole and ion trap operation modes by altering the pressure within the third vacuum chamber. The combination of the quadrupole and ion trap operation modes could provide new capabilities when solving analytical problems.  Based on the above overall situation, I think this manuscript can be published in this journal after minor revision.

Suggestions:

1.     An in-vacuum plasma was used in this paper, but few information of it has been provided, how about the geometry of the in-vacuum plasma, and what kind of drive power supply? And is there a rapid switching between two ion sources? The above information should be added at the appropriate place in the text.

2.     In page 8:”The mass spectrum of Perfluorotributylamine (PFTBA) covers a mass range from 20-600 Th, and some fragments could be observed, as shown in Figure 3c.” We only can find the obvious peaks from 31-502, the lighter ion of 20 and the ion of 600 can not be found in Figure 3C, please recheck this mass range. And the higher mass to charge ratios of 1000 Th in ion trap mode also should be corrected according to the actual detected m/z of 922.

3.     Why not use the same or adjacent scanning rate to characterize the resolution for ion trap mode and quadrupole mode? For example, both with 3200 Th/s scanning rate?

Comments on the Quality of English Language

1.     Some of mistake on spelling, such as ug/ml, R2 in the text.

Round 2

Reviewer 1 Report

Comments and Suggestions for Authors

All issues have been addressed properly. The manuscript is good for publication.